# Towards an Automatic Pollen Detection System in Ambient Air Using Scattering Functions in the Visible Domain

**DOI:** 10.3390/s22134984

**Published:** 2022-07-01

**Authors:** Jean-Baptiste Renard, Houssam El Azari, Jérôme Richard, Johann Lauthier, Jérémy Surcin

**Affiliations:** 1LPC2E-CNRS, 3A Avenue de la Recherche Scientifique, CEDEX 2, 45071 Orléans, France; houssam.elazari@lifyair.com (H.E.A.); jeremysurcin.pollutrack@protonmail.com (J.S.); 2LIFY-AIR, Le LAB’O, 1 Avenue du Champ de Mars, 45100 Orléans, France; jerome.richard@lifyair.com (J.R.); johann.lauthier@lifyair.com (J.L.)

**Keywords:** pollens, scattering, polarization, instrument

## Abstract

Pollen grains strongly affect human health by inducing allergies. Although the monitoring of airborne pollens particles is of major importance, the current measurement methods are manually conducted and are expensive, limiting the number of monitoring stations. Thus, there is a need for relatively low-cost instruments that can work automatically. The possible detection of pollen in urban ambient air (Paris, France) has been reported using the LOAC optical aerosol counter. These measurements indicate that the pollen grains and their nature could be determined using their scattering properties. For this purpose, the scattering functions (intensity and linear polarization) of 21 different airborne pollens were established in the laboratory using a PROGRA2 instrument. The linear polarization curves were close together, with a maximum polarization lower than 10% in the red domain and 5% in the green domain. The variability from one sample to another was partly due to the different sizes of the grains. An instrument with an absolute accuracy of about ±1% for polarization measurements should then be needed, coupled with a counting instrument to take into account the effects of size. On the other hand, the scattering curves for intensity presented with different shapes and strong differences up to a factor of 20 at some scattering angles, due to the size, shape, surface texture, and composition of the grains. Thus, we propose a proof of concept for new automated sensors that can be used in dense networks to count and identify pollen grains by analyzing the light they scatter at some specific angles.

## 1. Introduction

Pollen grains produced by wind-pollinated plants are responsible for allergies such as conjunctivitis, rhinitis, and asthma in millions of people around the world every year. At least 40% of the European population is affected by these allergies [1]. Not only do these allergies have a high impact on human health [2,3] but they also represent a significant financial burden on health care systems and on society as a whole [4]. Additionally, the prevalence of pollen-induced allergies is likely to increase and the symptoms are expected to become more severe due to climate change and atmospheric pollution [5,6]. Thus, monitoring airborne pollen grains is a task of major importance.

The first attempts at pollen monitoring date back to the end of the nineteenth century, with early aerobiologists designing the first pollen trapping experiments to investigate pollen’s role as a cause of allergic diseases [7]. The concept of collecting pollen grains on sticky microscope slides was born; it was followed by a series of pollen quantification experimental setups as described by [8], contributing to the development of volumetric samplers such as the one proposed by Hirst [9]. The measurement principle has not changed significantly since and is still currently widely used with manual analysis under a microscope to identify and quantify the pollen grains collected on the sticky slides. However, needs have changed, and this method is no longer tailored to allergy sufferers’ growing demand for accurate pollinic information. The analysis process is not only time-consuming and labor-intensive, but it is also costly as it requires dedicated qualified operators, making it impossible to have a dense monitoring network with sufficient spatial coverage. As an example, only 81 monitoring stations are available in Metropolitan France for a population of 65 million inhabitants and an area of 544,000 km^2^. Due to the need for manual processing, the pollen information is shifted in time by several days and does not represent the variabilities of pollen sources and the dissemination of particles well at a local scale.

To tentatively overcome these limitations, automated methods have been developed or are under development. Examples include techniques combining physical concepts such as infrared spectroscopy, Raman spectroscopy, fluorescent spectroscopy, and holography, and the use of machine learning models [10,11,12,13]. While providing promising results in real time, these techniques remain unable to address the cost and miniaturization constraints that are necessary for better spatial coverage and more localized pollen information (thousands of instruments would be necessary). Another physical concept could meet these constraints: light scattering in the visible domain. Some laboratories have shown that the light scattered by some pollens could be used to distinguish some pollen families from other ones—for example, by determining the scattering matrix in the backward scattering angle region and thus discriminating pollens by their polarization properties [14]. Additionally, inexpensive instruments relying on laser optics have been developed and used to classify some families of pollen grains by retrieving their forward and backward scattering intensities [15,16,17]. These optical techniques are very promising, but an extensive determination of the scattering properties of a large number of pollens is necessary to determine if pollens can indeed be easily identified in ambient air.

The possible detection of pollen grains in urban ambient air in the Park André Citroën in the south-west of Paris (France) has been reported by [18] using the LOAC optical aerosol counter [19], considering the time evolution of particles larger than 15 µm. This instrument has been operating almost continuously since 2013. As an example, Figure 1 presents the daily time-evolution for the year 2021 for all particles in the 0.2–30 µm size range (the presence of fog and rain has been removed). The authors have attributed the concentration increase of the largest particles of up to a factor of 10 between summer and winter to the presence of pollens, superimposed on a low concentration of background particles. In summer, the concentration of particles > 15 µm was a few hundreds of particles m^−3^, in agreement with typical pollen concentrations in ambient air [20].

Thanks to its light-scattering measurements performed at two different angles (~15° and ~60°), the LOAC can also provide an estimate of the typology of particles based on their ability to absorb light [18]. “Speciation zones” have been established in the laboratory with different families of particles that can be found in the ambient air and categorized as optically absorbing (carbonaceous particles), semi-transparent (mainly minerals), transparent (such as salts), and water droplets. Figure 2 presents a possible example of pollen detection (29 July 2021), characterized by a significant concentration of particles greater than 15 µm (top left) and their associated typology (top right). The typologies indicate particles between the optically absorbing and semi-transparent categories. As a comparison, the measurements at the end of the autumn (20 November 2021) showed concentrations up to 20 times lower; we can note the presence of carbonaceous particles for sizes <2 µm—as expected during the fall and winter seasons, originating especially from heating processes.

This possible pollen detection needed confirmation. The first step was to establish the light scattering curves of the pollen grains, to confirm that they indeed agreed with the LOAC detection. The second step was to propose a new optical device based on the optical properties of the pollens. For this purpose, we carried out a study on the light scattering curves of 21 pollens across almost the whole of the scattering angle domain. These measurements, which concerned the intensity and linear polarization of the pollens, were conducted in a laboratory for levitating pollen particles.

## 2. Data and Methods 

### 2.1. Laboratory Experimental Set-Up for the Scattering Curves of Pollens

The laboratory measurements of the optical scattering properties of pollens were conducted using an imaging gonio-photopolarimeter PROGRA2 instrument (French acronym “PRopriétés Optiques GRains Astronomiques et Atmosphériques”, for “optical properties of astronomical and atmospheric grains”). This provides the linear polarization and intensity scattering curves for a cloud of levitating particles with random orientations [21]. The version of PROGRA2 used for the present study performed measurements in the visible domain: PROGRA2-Vis [22]. The information and data obtained by the instrument for a large set of samples of different natures can be found at https://www.icare.univ-lille.fr/progra2/. Figure 3 presents the measurement concept of the PROGRA2-Vis instrument.

Pollens were sustained in levitation for several seconds using an air draught technique [23]. The particles crossed an unpolarized light source beam produced by a halogen white lamp, a depolarizer filter, and spectral filters (green domain 525–585 or red domain 620–680 nm). An optical fiber carried the light to the vial, in which the particles were lifted by a small air injection just before the measurements. A polarizing beam-splitter cube split the light scattered by the particles into its two components: parallel and perpendicular to the scattering plane (Iperp and Ipara). These fluxes were recorded by two synchronized cameras with similar fields of view. The use of these cameras allowed us to reject images where multiple scattering may have occurred and could thus bias the results (in general, multiple scattering reduces polarization). The resolution of the cameras was about 20 μm per pixel; therefore, only the largest pollen grains could be detected individually. Otherwise, the cameras recorded the scattered fluxes from the cloud of individual particles. 

The two detection cameras were mounted at fixed positions; the vial that contained the sample was mounted on a rotation device to vary the scattering angle in the 15–170° range between measurements. Measurements were conducted several times at a fixed angle to ensure that a good percentage of the number of images could be kept. The angles were changed by steps of 5° or 10°. A complete scattering function could be described by the measurements conducted over at least 19 different scattering angles.

The linear polarization P (in %) was retrieved from each pair of images recorded simultaneously by the Iperp and Ipara fluxes above a threshold level well above the electronic noise level, following the formula:P = 100 × (Iperp − Ipara)/(Iperp + Ipara)(1)

A third synchronized camera recorded the scattered light (Iref) at a constant scattering angle of 90°—thus rotating with the vial. This approach was motivated by the fact that the number of particles in the field of view can differ from one measurement to another. Thus, this third camera was used for normalization. The intensity I (in relative units) was retrieved after dividing the sum of fluxes recorded by the two first cameras by the flux of the third camera, following the formula:I = (Iperp + Ipara)/Iref(2)

Finally, all intensity scattering measurements were divided by measurements obtained in the 15–20° scattering angle range, to make a direct comparison of the amplitude of the curves possible without considering the size of the particles [24]. This procedure was also motivated by the fact that the scattered intensity at small scattering angles for irregular grains is almost independent of the shape and the refractive index of the particles and is only dependent on their size [19,25].

At least several tens of images are necessary in order to retrieve the mean scattering properties of the particles at a given scattering angle. This is mandatory to minimize the effect of the scattering intensity variability produced by individual particles [26] when they cross the field view of the detectors.

Depending on their color, some pollens cannot be studied in the green domain because the sensitivity of the PROGRA2 cameras is lower in this domain than in the red domain, and so cannot allow us to obtain usable images.

### 2.2. Samples

A total of 21 different pollen species were studied in the red spectral domain and 8 in the green spectral domain. The samples were collected and provided by the Stallergenes Greer Company (in the US). Some of them are the most allergenic pollens. Their size measurements were given by the provider (Table 1).

Although pollens appear more or less spherical, their surface is not smooth. For example, the surface of ragweed pollen is spiky, the surface of olive tree pollen looks like a sponge, and the surface of mugwort exhibits some kinds of lobes. Additionally, depending on their ability to sustain or not sustain their hydration levels (wet or dry particles), their shape can evolve from a rough spherical shape to more compact grains with a crumpled appearance. Thus, the scattering properties of pollens will differ from the Mie scattering calculations for a perfect sphere, and potential attempts at modelling them could be complicated—although, some promising results have been obtained for pollens with perfect regular shapes [27]. Thus, we can expect some differences from one species to another.

In the following, we considered three main pollen categories: grass, weeds, and tree pollens (grass and weed pollen scattering curves are grouped together in the same figures while tree pollen scattering curves are on separate figures in the red spectral domain). Their sizes ranged from 15 µm to 100 µm, although most of them remained in the 20–35 µm domain. Nevertheless, the pollens’ growth, and hence their sizes, could differ from one location to another, depending on the weather conditions and the subsoils. Therefore, the size values used below to tentatively establish a link between size and scattering properties could differ from freshly emitted pollen in the ambient air.

## 3. Results

### 3.1. Polarization Curves

The evolution of the polarization with scattering angles for the 21 samples is presented in Figure 4 and Figure 5 for the red spectral domains. The values for the maximum polarization—which occurred for angles in the 80–100° range—were relatively low, in the 5–15% range. Some oscillations in the curves that occurred for a few samples at large scattering angles could have been due to the particle’s shape and surface irregularities. 

The polarization was slightly lower in the green domain (Figure 6, with a maximum polarization in the 0–7% range, close to the 8 ± 4% maximum polarization value for cypress pollen obtained by [28]. Lower polarization values in the green domain compared to the red domain were often observed, in agreement with previously retrieved polarization curves for most large mineral particles [29,30]. 

The values for maximum polarization often increase with the size of particles to reach a kind of saturation value that depends on the particle’s nature [31]. When combining maximum polarization values for the 21 pollen samples in the red domain and the 8 samples in the green domain, such tendencies could be established (Figure 7) despite the dispersion coming from the different compositions and shapes of the particles. In the red domain, the trend could be established using a second order polynomial, showing the possible polarization saturation at about 11 ± 2% for sizes greater than 60 µm. The trend was more difficult to establish in the green domain (a linear fit is used in the figure); it was, however, similar to the one in the red domain. The size effect produced an increase in polarization of about a factor of 2 in the 15–100 µm pollen size range. Thus, the differences between the different polarization curves in Figure 4, Figure 5 and Figure 6 were driven both by the size of the pollens and by the sources of emission.

The polarization curves of the pollens surprisingly looked like those of sand [30,32]. Considering the measurements error bars, it could be difficult (but not impossible) to unambiguously identify the presence of pollens in ambient air and to estimate their nature with polarization measurements only, whatever the wavelength is. An instrument with an absolute accuracy of about ±1% for polarization measurements could be needed, coupled with a counting instrument to consider the size effect.

### 3.2. Intensity Curves

The scattering curves were greatly more sensitive to the nature of pollens (Figure 8 and Figure 9) than the polarization curves. Variations of up to a factor of 20 occurred between the different samples for scattering angles larger than 60°. Such variations were due to the size of the particles (in general, curves should be flatter when the size of the particles increases), but also due to the composition and thus the refractive index, shape, and surface texture of the pollens. No obvious relationship between all these parameters could be proposed here because of the diversity of the pollens’ aspects, requiring complex modeling calculations that could be conducted in the future. Nevertheless, some curves looked like the theoretical curves obtained for spheres with hexagonal grids and barbs, with an amplitude of around 10 between 15° and ~120° [27].

The curve for the ragweed exhibited peculiar behavior, with an almost constant intensity between the 15 and 60° scattering angles, while the intensities strongly decreased for the other samples. This could have resulted from the spiky surface of the sample, as shown in Figure 10 [33]. The curve for the fir tree pollen was almost flat, which could be explained by the large size of the particles in the 50–100 µm range (Figure 10), as already observed with PROGRA2 for large mineral and salt grains (Figure 11). The curve for the olive tree pollen slightly decreased with increasing angles; this could be due to its peculiar rough surface with a spongy aspect (Figure 9). For the other samples, two families seemed to be present: pollens with a curve amplitude smaller than a factor of 10 (such as for the birch and vernal grass pollens) and pollens with a curve amplitude greater than a factor of 10 (such as for the Mugwort). Among these families, some differences were also detected—sometimes with an increase in intensity for the largest scattering angles. Similar results were obtained for measurements in the green domain, without any significant differences from the red domain; thus, we will only consider the red domain measurements here.

### 3.3. Discussion

With the objective of identifying pollens in ambient air, it is necessary to compare these scattering curves to those of other families of particles greater than 15 µm that can be found in the ambient air. Aggregated and/or compact carbonaceous particles (black carbon, organic carbon, soot) that are optically absorbing, and mineral particles (sands, cement, basalt) that are transparent and semi-transparent, have been also studied with PROGRA2. These samples are similar to those used to establish the LOAC speciation zones in Figure 2. The individual curves were averaged to produce the “carbonaceous” and “mineral” curves, and the error bars correspond to the error of the mean (Figure 11). 

Most of the pollen curves were in-between the more carbonaceous particles and the mineral particles (Figure 11), confirming what has been observed with the LOAC instrument for the scattering angle of ~60°. Obviously, a confusion between pollens and pollution particles could occur in case of large aggregates composed of a mixture of carbonaceous and non-carbonaceous monomers. Nevertheless, the pollen PROGRA2 curves also show some specific evolution at large scattering angles, with or without intensity increases with increasing angles. 

The concentration of pollens during the pollen season can reach several hundred particles m^−3^ and even more than 1000 particles m^−3^ [20]. This is well above the typical concentrations of less than 100 particles m^−3^ with a size >15 µm during urban pollution events not in the vicinity of building activities or coal-user industries [18]. Therefore, the presence of pollens in ambient air could probably be unambiguously detected by a dedicated optical counting device performing measurements at several scattering angles most of the time.

The present study has shown that performing measurements of the pollens’ scattering properties at several angles in the 15–160° range in ambient air, combined with their size measurements obtained from observations at a small scattering angle (LOAC concept), could be an original and promising approach to detect pollens in real time and to estimate their origin. Nevertheless, such an approach based on laboratory measurements needs to be validated with real measurements in ambient air and to be compared to real measurements performed routinely by the networks using Hirst sensors [8,9].

This will be the objective of a new instrument, combining the LOAC concept [19] for counting to determine the size and concentration of the particles in real time, and to perform typology determination at several specific angles based on these PROGRA2 results. Such an instrument must be light and substantially cheaper compared to other instruments that perform real-time detection of pollens [10,11,12,13] in order to be deployed in dense networks. Such networks could allow for the better analysis of the heterogeneity of pollen emissions, concentrations, and transport.

This new, light, and relatively low-cost instrument has already been developed and tested in the laboratory, with the aim of being able to recognize different pollen taxa. Some versions of it are now deployed outdoors in networks at various sites in France to conduct a validation campaign. The measurement principles of this new sensor as well as the first results will be presented in an upcoming paper that is expected to be released before the end of 2022.

## 4. Conclusions

Pollens exhibit intensity scattering curves that could allow us to detect them among other particles in ambient air and to tentatively distinguish between their various families. The polarization curves presented some differences of a few % from one sample to another, but the variations were weak and were partly due to the pollens’ sizes. On the other hand, intensity curves often differed from one pollen to another—although some curves showed similarities in some scattering angle regions. These differences were due to the composition, shape, and surface texture of the particles. 

These principles on the optical properties of pollens were applied to the development of a new light instrument to monitor in real time the pollens’ concentrations and their typologies. The first results and the validation of such an instrument will be presented in the following months.

## Figures and Tables

**Figure 1 sensors-22-04984-f001:**
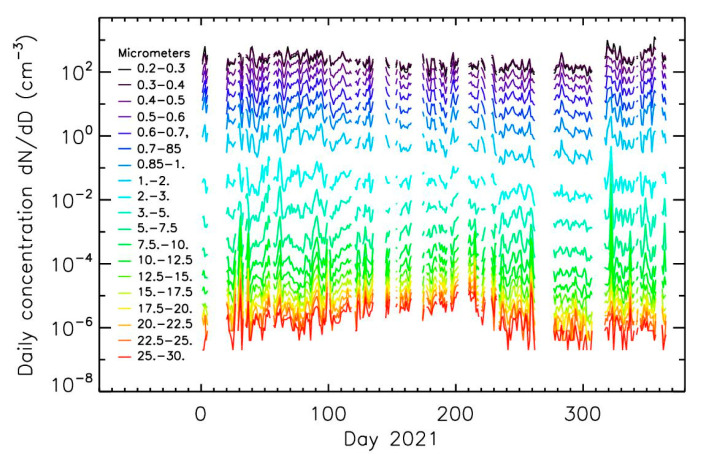
Daily time-evolution of particle concentrations for the 19 size classes measured by the LOAC optical counter in 2021 in the Park André Citroën in the south-west of Paris (France); pollen grains can be present in size classes > 15 µm.

**Figure 2 sensors-22-04984-f002:**
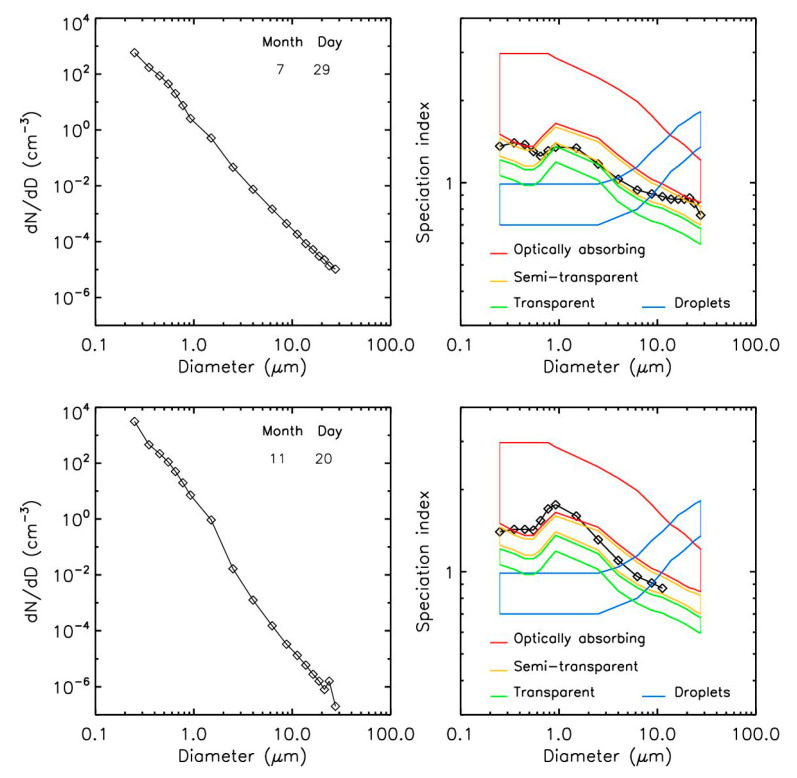
Size distribution (**left**) and typology (**right**) from the LOAC measurements; (**top**) during summer, (**bottom**) during late in the fall.

**Figure 3 sensors-22-04984-f003:**
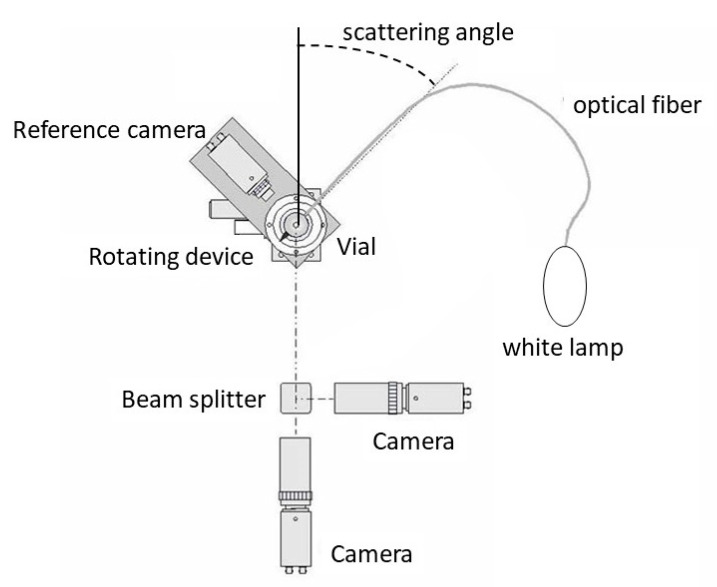
Concept of measurement with the PROGRA2 instrument.

**Figure 4 sensors-22-04984-f004:**
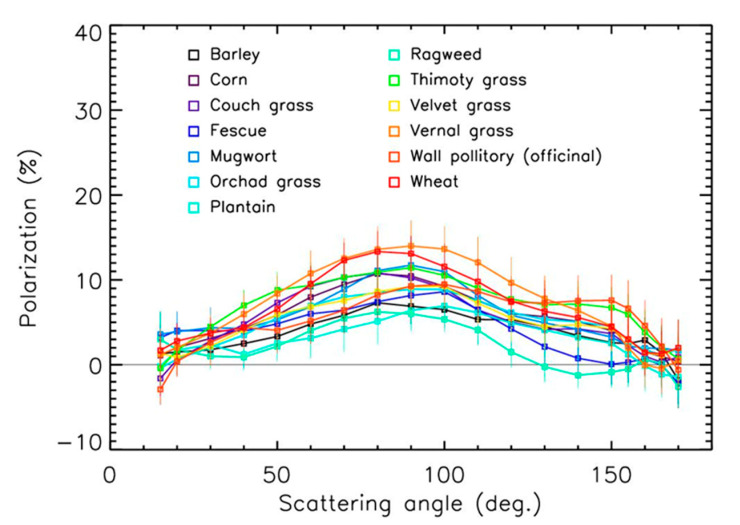
Polarization curves in the red domain for grass and weed pollens.

**Figure 5 sensors-22-04984-f005:**
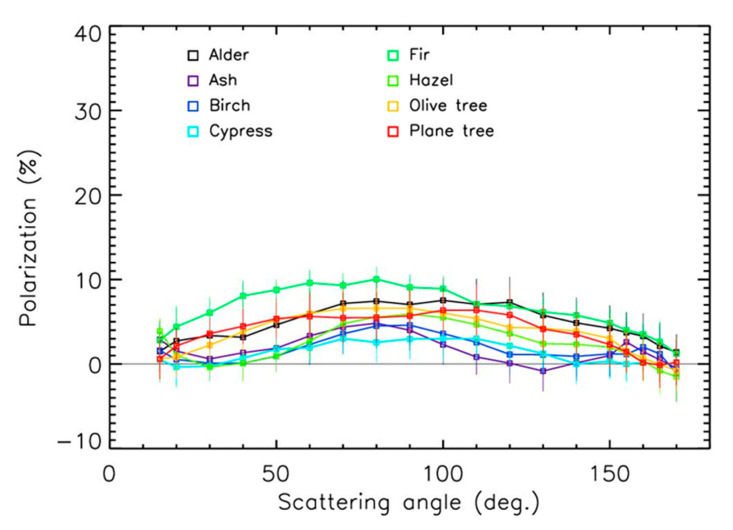
Polarization curves in the red domain for tree pollens.

**Figure 6 sensors-22-04984-f006:**
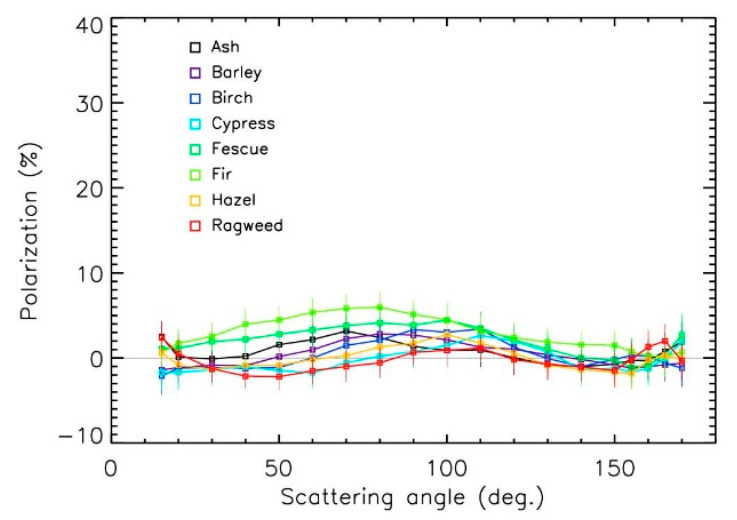
Polarization curves in the green domain.

**Figure 7 sensors-22-04984-f007:**
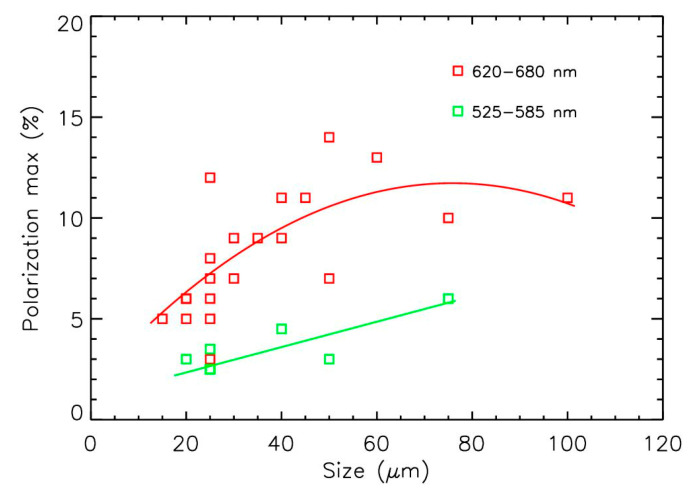
Evolution of the maximum value of polarization with the size of the pollens.

**Figure 8 sensors-22-04984-f008:**
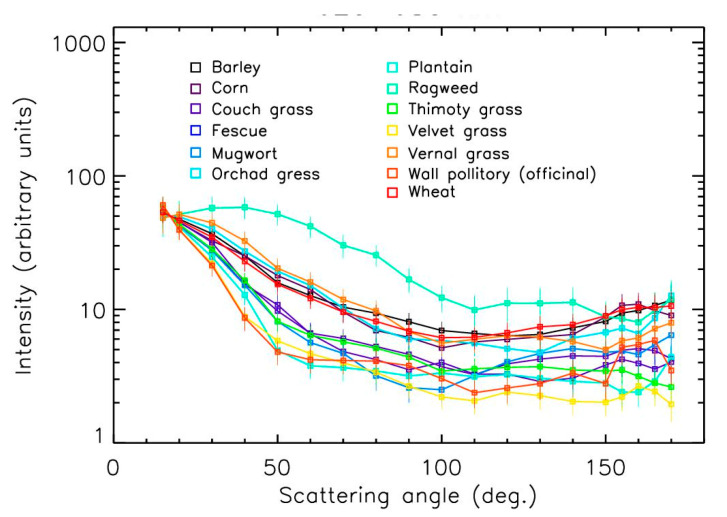
Intensity curves in the red domain for grass and weed pollens.

**Figure 9 sensors-22-04984-f009:**
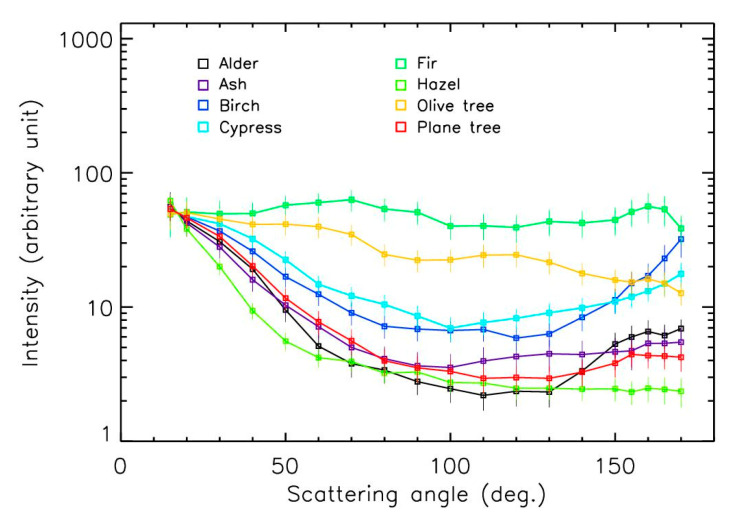
Intensity curves in the red domain for tree pollens.

**Figure 10 sensors-22-04984-f010:**
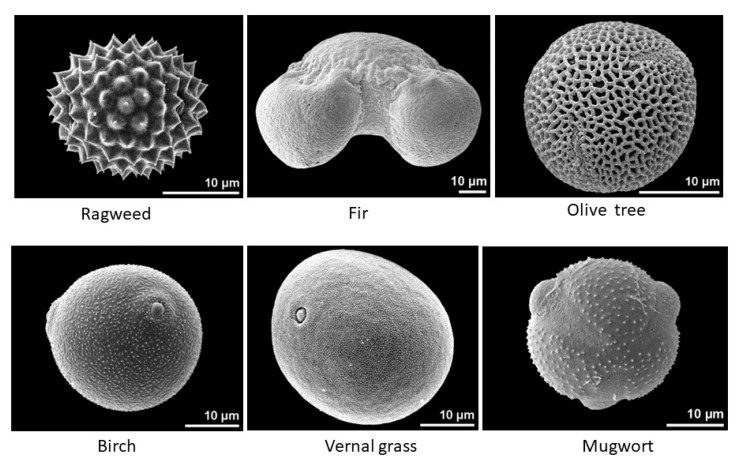
Images of some pollens, showing the diversity in their size and surface shape; images from the PalDat—Palynological Database [33].

**Figure 11 sensors-22-04984-f011:**
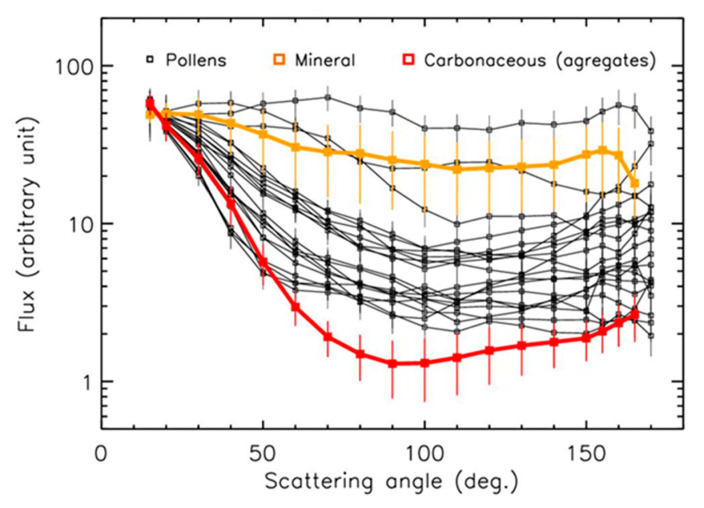
Intensity curves in the red domain for the pollens compared to various mineral and carbonaceous particles.

**Table 1 sensors-22-04984-t001:** Pollen species and sizes (the sizes were rounded to the closest 5 µm).

Family	Common Name	Size (µm)
Grass	Barley	50
	Corn	100
	Couch grass	45
	Fescue	40
	Orchard grass	35
	Timothy grass	40
	Velvet grass	30
	Vernal grass	50
	Wheat	60
Weed	Mugwort	25
	Plantain	30
	Ragweed	20
	Wall pellitory	15
Tree	Alder	25
	Ash	20
	Birch	25
	Cypress	25
	Fir	75
	Hazel	25
	Olive tree	25
	Plane tree	20

## Data Availability

The data will be available at https://www.icare.univ-lille.fr/progra2-en/banque-de-donnees/.

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
