# Peer review of "Towards an Automatic Pollen Detection System in Ambient Air Using Scattering Functions in the Visible Domain"

_sensors, 2022, doi:10.3390/s22134984_

Round 1

Reviewer 1 Report

The authors analyse the light scattered by pollen grains at some specific angles to count and identify the pollen taxons. The paper is clear and easy to follow, however the main weaknesses of the manuscript is poor discussion of the results with other studies and lack of the verification for a real case.

It has to be clearly declared what is the added value of this study compared to other papers that uses optical approaches and has been published in recent years. In my opinion a detection possibilities in a “real case” with a mixture of pollen grains should be conducted and compared with a standard measurements (Hirst sampler) before the considering the paper to the publication.

The structure of the manuscript should be corrected. The following sections should be included: Introduction, Data and Methods, Results, Discussion (alternatively Results and discussion), Conclusions

Author Response

We would like to thank the reviewers for their constructive comments. They have helped us to improve the paper.

Reviewer 1

Comments: The authors analyse the light scattered by pollen grains at some specific angles to count and identify the pollen taxons. The paper is clear and easy to follow, however the main weaknesses of the manuscript is poor discussion of the results with other studies and lack of the verification for a real case.

Our answer: We understand the reviewer concern. To our knowledge, there are no other measurements of the whole scattering functions of pollens except for cypress given by Gómez Martín et al. (reference 26) and discussed in line 202. Thus, it is difficult to discuss our laboratory results compared to other studies. On the other hand, the verification for a real case will be the object of an upcoming paper that will be submitted in Sensors before the end of 2022.

Comments: It has to be clearly declared what is the added value of this study compared to other papers that uses optical approaches and has been published in recent years. In my opinion a detection possibilities in a “real case” with a mixture of pollen grains should be conducted and compared with a standard measurements (Hirst sampler) before the considering the paper to the publication.

Our answer: We have added in the discussion (now part 3.3):

“Nevertheless, such approach based on laboratory measurements needs to be validated with real measurements in ambient air and to be compared to real measurements performed in routine by the networks using Hirst sensors [8,9].

This will be the objective of a new instrument, combining the LOAC concept [19] for counting, to determine the size and the concentration of the particles in real time, and to perform typology determination at several specific angles based on these PROGRA2 results. Such an instrument must be light and strongly much cheaper compared to other instruments that perform real-time detection of pollens [10-13], in order to be deployed in dense networks. Such networks could allow to better analyze the heterogeneity of pollens emissions, concentrations, and transport.

This new light and relatively low-cost instrument has already been developed and tested in laboratory, with the aim of being able to recognize different pollen taxa. Some versions of it are now deployed outdoor in network at various sites in France, to conduct a validation campaign. The measurement principles of this new sensor as well as the first results will be presented in an up-coming paper expected to be released before the end of 2022.”

We have also changed the end of the conclusion to:

“These principles on the optical properties of pollens are applied to the development of a new light instrument to monitor in real time the pollens’ concentrations and their typologies. The first results and the validation of such an instrument will be presented in the followings months.”

Comments: The structure of the manuscript should be corrected. The following sections should be included: Introduction, Data and Methods, Results, Discussion (alternatively Results and discussion), Conclusions

Our answer: We have revised the structure of the paper according to the reviewer suggestion.

Reviewer 2 Report

Pollen grains are not only an important and typical atmospheric aerosol component that influences weather and climate, but also on their effects on human health by inducing allergies. However, accurate detection of pollens is not simple. This study introduces a new method to detect pollen particles in ambient air using their scattering functions in the visible domain. This is an important piece of work that could have great impact, and the manuscript is well organized and presented. I have only a few minor suggestions that should be noticed by the authors to further improve the manuscript.

1. Some meaningful reviews on pollens and its monitoring are suggested to be discussed to improve the introduction session. For example, https://doi.org/10.1021/ac801791a and DOI 10.1007/s00334-010-0261-3.

2. Y-axis label of Figure 1 is incorrect, it should be “concentration” instead of “oncentration”.

3. A schematic figure to illustrate the framework of the instrument is suggested to better introduce the measurements, and more quantitative information on the measurements would also be useful for further applications related to the measurements.  

4. As the images shown in Figure 9, pollen grains should quite non-spherical but kind of “regular” shapes that could be considered in numerical models (https://doi.org/10.1364/OE.24.00A104). As I noticed, there was a comprehensive numerical model for pollen scattering properties by representing complex geometries of morning glory pollens. Similar works would be important counterpart numerical works support this observational work. Thus, such numerical studies are suggested to be discussed in the work.

5. How is the quality control performed in the study? 

6. Is there any plan to further use those observations quantitatively? 

Author Response

We would like to thank the reviewers for their constructive comments. They have helped us to improve the paper.

Reviewer 2

Pollen grains are not only an important and typical atmospheric aerosol component that influences weather and climate, but also on their effects on human health by inducing allergies. However, accurate detection of pollens is not simple. This study introduces a new method to detect pollen particles in ambient air using their scattering functions in the visible domain. This is an important piece of work that could have great impact, and the manuscript is well organized and presented. I have only a few minor suggestions that should be noticed by the authors to further improve the manuscript.

  1. Some meaningful reviews on pollens and its monitoring are suggested to be discussed to improve the introduction session. For example, https://doi.org/10.1021/ac801791a and DOI 10.1007/s00334-010-0261-3.

Our answer: We have added both proposed references to enrich the introduction section and we have changed the text to: “The first attempts of pollen monitoring date back to the end of the nineteenth century, with early aerobiologists designing the first pollen trapping experiments to investigate the pollens’ role as a cause of allergic diseases [7]. The concept of collecting pollen grains on sticky microscope slides was born. It was followed by a series of pollen quantification experimental setups as described by [8], contributing to the development of volumetric samplers as the one proposed by Hirst [9]. The measurement principle has not changed significantly since and still currently widely used with a manual analysis under microscope to identify and quantify the pollen grains impacted onto the sticky slides. However, the needs have changed, and this method is no longer tailored to the allergy sufferers growing demand of an accurate pollinic information.”

  1. Y-axis label of Figure 1 is incorrect, it should be “concentration” instead of “oncentration”.

Our answer: Sorry for the mistake. We have corrected the figure.

  1. A schematic figure to illustrate the framework of the instrument is suggested to better introduce the measurements, and more quantitative information on the measurements would also be useful for further applications related to the measurements.  

Our answer: We have added a new figure (now figure 3) showing the diagram of PROGRA2-Vis. We have also added: “At least several tens of images are necessary to be able to retrieve the mean scattering properties of the particles at a given scattering angle. This is mandatory to minimize the effect of the scattering intensity variability produced by individual particles [26] when they cross the field view of the detectors.”

We have added the reference “ Renard, J.-B.; Geffrin, J.-M.; Tobon Valencia, V.; Tortel, H.; Ménard, F.; Rannou, P.; Milli, J.; Berthet, G. Number of independent measurements required to obtain reliable mean scattering properties of irregular particles having a small size parameter, using microwave analogy measurements, J Quant Spectrosc Radiat Transfer 2021; 272; 107718.”

  1. As the images shown in Figure 9, pollen grains should quite non-spherical but kind of “regular” shapes that could be considered in numerical models (https://doi.org/10.1364/OE.24.00A104). As I noticed, there was a comprehensive numerical model for pollen scattering properties by representing complex geometries of morning glory pollens. Similar works would be important counterpart numerical works support this observational work. Thus, such numerical studies are suggested to be discussed in the work.

Our answer: We have added, in the part 2.2: ...” although some promising results were obtained when the pollens have perfect regular shapes [27].”. We have also added in part 3.2: “Nevertheless, some curves look like the theoretical curves obtained for spheres with hexagonal grids and barbs, with an amplitude around 10 between 15° and ~120° [27].”

We have also added the reference:” Liu, C.; Yin, Y. Inherent optical properties of pollen particles: a case study for the morning glory pollen. Optics Express 2016; 24(2); A104-A113.”

  1. How is the quality control performed in the study? 

Our answer: If the reviewer means the quality control of PROGRA2, the instrument performances and optical alignment are controlled before the sessions of measurements. If the reviewer means the quality control of the ability of a new instrument to detect and identify the pollens, we think that we have answered below.

  1. Is there any plan to further use those observations quantitatively? 

Our answer: We have added in the discussion (now part 3.3):

“Nevertheless, such approach based on laboratory measurements needs to be validated with real measurements in ambient air and to be compared to real measurements performed in routine by the networks using Hirst sensors [8,9].

This will be the objective of a new instrument, combining the LOAC concept [19] for counting, to determine the size and the concentration of the particles in real time, and to perform typology determination at several specific angles based on these PROGRA2 results. Such an instrument must be light and strongly much cheaper compared to other instruments that perform real-time detection of pollens [10-13], in order to be deployed in dense networks. Such networks could allow to better analyze the heterogeneity of pollens emissions, concentrations, and transport.

This new light and relatively low-cost instrument has already been developed and tested in laboratory, with the aim of being able to recognize different pollen taxa. Some versions of it are now deployed outdoor in network at various sites in France, to conduct a validation campaign. The measurement principles of this new sensor as well as the first results will be presented in an up-coming paper expected to be released before the end of 2022.”

We have also changed the end of the conclusion to:

“These principles on the optical properties of pollens are applied to the development of a new light instrument to monitor in real time the pollens’ concentrations and their typologies. The first results and the validation of such an instrument will be presented in the followings months.”

Round 2

Reviewer 1 Report

The manuscript has been improved. The authors have not included the evaluation of the results but I think the paper can be published in the current form.